# Development of Coupled Biokinetic and Thermal Model to Optimize Cold-Water Microbial Enhanced Oil Recovery (MEOR) in Homogenous Reservoir

**Eunji Hong, Moon Sik Jeong, Tae Hong Kim, Ji Ho Lee, Jin Hyung Cho and Kun Sang Lee ***

Department of Earth Resources and Environmental Engineering, Hanyang University, Seoul 04763, Korea;
ejihong@hanyang.ac.kr (E.H.); qlrhkd0507@hanyang.ac.kr (M.S.J.); kth1014@hanyang.ac.kr (T.H.K.);
shri0112@hanyang.ac.kr (J.H.L.); petrocho87@hanyang.ac.kr (J.H.C.)
* Correspondence: kunslee@hanyang.ac.kr; Tel.: +82-2-2220-4427

**Abstract:** By incorporating a temperature-dependent biokinetic and thermal model, the novel method, cold-water microbial enhanced oil recovery (MEOR), was developed under nonisothermal conditions. The suggested model characterized the growth for *Bacillus subtilis* (microbe) and *Surfactin* (biosurfactant) that were calibrated and confirmed against the experimental results. Several biokinetic parameters were obtained within approximately a 2% error using the cardinal temperature model and experimental results. According to the obtained parameters, the examination was conducted with several injection scenarios for a high-temperature reservoir of 71 °C. The results proposed the influences of injection factors including nutrient concentration, rate, and temperature. Higher nutrient concentrations resulted in decreased interfacial tension by producing *Surfactin*. On the other hand, injection rate and temperature changed growth condition for *Bacillus subtilis*. An optimal value of injection rate suggested that it affected not only heat transfer but also nutrient residence time. Injection temperature led to optimum reservoir condition for *Surfactin* production, thereby reducing interfacial tension. Through the optimization process, the determined optimal injection design improved oil recovery up to 53% which is 8% higher than waterflooding. The proposed optimal injection design was an injection sucrose concentration of 100 g/L, a rate of 7 m$^3$/d, and a temperature of 19 °C.

**Keywords:** microbial enhanced oil recovery (MEOR); biokinetics; biosurfactant; *Bacillus subtilis*; thermal modeling; high temperature reservoir

## 1. Introduction

Microbial enhanced oil recovery (MEOR) is a promising technique as an economic and environmentally friendly approach [1,2]. MEOR recovers additional oil from a reservoir using microbes and their by-products. It is similar to chemically enhanced oil recovery (CEOR) in a manner that uses surfactants or polymers [3]. However, there are pronounced differences as the used products are affordable as well as biodegradable. Previous studies have reported that MEOR is sufficient to improve oil recovery [4,5].

A number of problems are associated with MEOR because the amount of byproducts is influenced by growth conditions such as pH, salinity, and temperature [6,7]. Therefore, these effects should be quantified using an appropriate kinetic model for a specific microorganism [8]. Although quantification plays a key role, most studies have not considered it [9–16]. The process is too complex and difficult because of the series of microbial activities during MEOR. Thus, numerical modeling of MEOR considering the geoenvironmental effects remains a challenge.

Among the factors of growth, reservoir temperature is far more important than other factors because most microbes do not grow at temperatures higher than 80 °C [17]. Microbial activity is not able to be sustained outside of a survival temperature range. Even with thermophilic microbes, the effect is small because the microbial growth rate has parabolic relationship with temperature. When the temperature condition is lower or higher than the optimal growth temperature, microbial growth activity decreases because the microbial death rate is higher than the growth rate [18,19]. Therefore, temperature is the key factor for suitable MEOR application in a high-temperature reservoir.

Only a few studies have quantified the effects of growth conditions on MEOR performance. Vilcáez et al. [20] investigated the effects of reservoir pH conditions on MEOR under an isothermal condition. However, their model was limited because temperature was of greater influence than pH in a high-temperature reservoir. Using *Clostridium* sp., an experimental study was conducted to identify the possibility of MEOR in a high-temperature reservoir [21]. Using numerical modeling, Sivasankar and Kumar [22] and Hosseininoosheri et al. [7] investigated the effects of in-situ reservoir temperature on MEOR performance. However, the coupled effect of injection temperature, rate, and nutrient concentration was not examined. Consideration of these effects is crucial because a change in injection design can affect MEOR efficiency according to microbial activity in high-temperature reservoirs. Only a lumped microbial reaction has been utilized for numerical modeling of MEOR. Furthermore, to the best our knowledge, it was the first attempt to generate a stoichiometric equation and optimize injection design with thermally active biosurfactant. Using a combined stoichiometric equation, biokinetics, and thermal model, this study implemented a parametric analysis and an optimization process under nonisothermal conditions to assess cold-water MEOR performance.

Cold-water MEOR was developed to overcome the limitations of MEOR-applicable temperatures. The effect of temperature on reaction rates was formulated with a multi-activation energy table. Stoichiometric equations and reaction rates were obtained via history-matching with laboratory experiments. We assumed that *Surfactin* was a dominant byproduct of *Bacillus Subtilis* and ignored the adsorption of any component to analyze only the effect of interfacial tension by *Surfactin*. In addition, partitioning of the produced *Surfactin* was not considered to be simplify the model. Based on changes in relative permeability via a reduction in interfacial tension, a wettability modification by*Surfactin* was established. Finally, the performance of cold-water MEOR was evaluated in a high-temperature reservoir. Injection nutrient concentration, rate, and temperature were considered in a parametric analysis of a successful cold-water MEOR application. Each injection parameter was independently adjusted to examine the effects on oil recovery. With an objective function to maximize oil recovery, the optimal injection design was also constructed by simultaneously changing the injection parameters. The overall processes of numerical modeling and history-matching were conducted using CMG STARS (2017) and CMG CMOST (2017), respectively [23,24].

## 2. Materials and Methods

### 2.1. Stoichiometric Reactions of Microbial Growth and Biosurfactant Production

Microbial reactions were formulated based on the method outlined by Rittmann and McCarty [25]. According to the method, microbial growth depends on energy production ($R_e$) and cellular synthesis ($R_s$). The half-reactions of the acceptor, cell, and donor were designated as $R_a$, $R_c$, and $R_d$, respectively. The two basic reactions are as follows:

$$R_e = R_a - R_d \tag{1}$$

$$R_s = R_c - R_d \tag{2}$$

The overall reaction was obtained from knowledge of the fraction as follows:

$$R = f_e R_e - f_s R_s \tag{3}$$

where $f_e$ and $f_s$ are the fractions used for energy and cell synthesis or production, respectively. The sum of these partitions is 1 [25].

For the acceptor, an electron equivalent ($e_i$) should be used for the energy production balance (Equation (4)) as follows:

$$R_e = \sum_{i=1}^{n} e_i R_a - R_d \tag{4}$$

where $e_i$ is electron equivalents [-eq/mol], and $n$ is number of component [dimensionless].

The half reaction of energy production was formulated using electron equivalents as shown in Table 1 [20]. Equations (5)–(7) list the half reactions for energy production of sucrose and fermentation [25].

$$
\begin{aligned}
&0.02083 \text{ Sucrose} + 0.02350 \text{ HCO}_3^- \rightarrow \\
&0.02346 \text{ Fructose} + 0.01175 \text{ Lactate} + 0.01175 \text{ Acetate} + 0.00235 \text{ Mannitol} \\
&+0.01175 \text{ Ethanol} + 0.03642 \text{ CO}_2 + 0.00133 \text{ H}_2\text{O}
\end{aligned} \tag{5}
$$

$$
\begin{aligned}
&0.02083 \text{ Sucrose} + 0.05 \text{ NH}_4^+ + 0.05 \text{ HCO}_3^- \rightarrow \\
&0.05 \text{ } Bacillus \text{ } S \text{ } ubtilis + 0.05 \text{ CO}_2 + 0.17917 \text{ H}_2\text{O}
\end{aligned} \tag{6}
$$

$$
\begin{aligned}
&0.02083 \text{ Sucrose} + 0.00362 \text{NH}_4^+ + 0.00362 \text{HCO}_3^- \rightarrow \\
&0.00362 Surfactin + 0.06159 \text{CO}_2 + 0.06975 \text{H}_2\text{O}
\end{aligned} \tag{7}
$$

**Table 1.** Electron equivalents.

| Electron Acceptor | Moles | $e$-eq/mol | Moles $\times$ $e$-eq/mol | $e_i$ = (Moles $\times$ $e$-eq/mol)/sum |
|---|---|---|---|---|
| Lactate | 0.5 | 12 | 6 | 0.141 |
| Acetate | 0.5 | 8 | 4 | 0.094 |
| Mannitol | 0.1 | 26 | 2.6 | 0.0610 |
| Ethanol | 0.5 | 12 | 6 | 0.141 |
| Fructose | 1 | 24 | 24 | 0.563 |
| | | | Sum = 42.6 | |

With the electron equivalents and half reactions, stoichiometric equations were obtained as shown in Equations (8) and (9).

$$
\begin{aligned}
&0.02083 \text{ Sucrose} + 0.05 \text{ } f_e\text{NH}_4^+ + (0.05f_s + 0.02350f_e)\text{HCO}_3^- \xrightarrow{Bacillus\ subtilis} \\
&0.05f_s Bacillus \text{ } Subtilis + 0.02346 f_e\text{Fructose} + 0.01175f_e\text{Lactate} + 0.01175f_e \text{ Acetate} \\
&+0.00235 f_e\text{Mannitol} + 0.01175 f_e\text{Ethanol} + (0.05f_s + 0.03642f_e) \text{ CO}_2 \\
&+(0.17917f_s + 0.00133f_e) \text{ H}_2\text{O}
\end{aligned} \tag{8}
$$

$$
\begin{aligned}
&0.02083 \text{ Sucrose} + 0.00362 \text{ } f_e'\text{NH}_4^+ + (0.00362f_s' + 0.2350f_e')\text{HCO}_3^- \xrightarrow{Bacillus\ subtilis} \\
&0.00362f_s' Surfactin + 0.02346f_e'\text{Fructose} + 0.01175 \text{ } f_e'\text{Lactate} + 0.01175f_e'\text{Acetate} \\
&+0.00235 \text{ } f_e'\text{Mannitol} + 0.01175 \text{ } f_e'\text{Ethanol} + (0.06159f_s' + 0.03642f_e') \text{ CO}_2 \\
&+(0.06975f_s' + 0.00133f_e') \text{ H}_2\text{O}
\end{aligned} \tag{9}
$$

where $f_s$ is a synthesis fraction for *Bacillus subtilis* [dimensionless], $f_e$ is an energy fraction for *Bacillus subtilis* [dimensionless], $f_s'$ is a synthesis fraction for *Surfactin* [dimensionless], and $f_e'$ is an energy fraction for *Surfactin* [dimensionless].

These overall microbial reactions represent the synthesis of microbial cells and bioproducts. Using the following mass balance (Equation (10)), the fractions from the sucrose used for *Bacillus Subtilis* synthesis ($f_s$) and *Surfactin* production ($f_s'$) were determined against experimental data.

$$\sum n_{Reac,i} M_{Reac,i} = \sum n_{Prod,i} M_{Prod,i} \tag{10}$$

where $n_{\text{Reac},i}$ is number of moles of reactant $i$ [dimensionles], $M_{\text{Reac},i}$ is molecular weight of reactant $i$ [kg/m$^3$], $n_{\text{Prod},i}$ is number of moles of product $i$ [dimensionless], and $M_{\text{Prod},i}$ is molecular weight of product $i$ [kg/m$^3$].

## 2.2. Nonisothermal Biokinetics

The Monod equation is the most commonly used kinetic model of microbial growth and metabolism. Monod kinetics can formulate the growth of all products and cells [26,27]. The basic form of the Monod equation was developed using the microbial growth rate as follows (Equation (11)).

$$r_g = \frac{r_{\max}S}{K_s + S} \tag{11}$$

where $r_g$ is growth rate [1/d], $r_{\max}$ is maximum growth rate [1/d], $S$ is nutrient mass concentration [g/L], $K_s$ is nutrient concentration representing at which $r_g$ is $r_{\max}/2$ [g/L].

With microbial growth rate ($r_g$), the change in the microbial mass concentration can be expressed as follows (Equation (12)).

$$\frac{dX}{dt} = r_g X \tag{12}$$

where $X$ is microbial mass concentration [g/L].

To describe the effect of temperature on biokinetics, reaction rate was described as a function of temperature as shown in Equation (13) as follows:

$$r_{\max} = r_{\text{opt}} f(T) \tag{13}$$

where $r_{\text{opt}}$ is optimal reaction rate at optimal temperature [1/d].

Rosso et al. [28] formulated an equation for $f(T)$ with $T_{\max}$, $T_{\min}$, and $T_{\text{opt}}$ as shown in Equation (14). These temperatures can be obtained from the experimental data.

$$f(T) = \frac{(T - T_{\max})(T - T_{\min})^2}{(T_{\text{opt}} - T_{\min})\left[(T_{\text{opt}} - T_{\min})(T - T_{\text{opt}}) - (T_{\text{opt}} - T_{\max})(T_{\text{opt}} + T_{\min} - 2T)\right]} \tag{14}$$

where $f(T)$ is cardinal temperature model of Rosso et al. [28] [dimensionless], $T$ is present temperature environment [K], $T_{\max}$ is maximum survival temperature [K], $T_{\min}$ is minimum survival temperature [K], and $T_{\text{opt}}$ is optimum survival temperature [K].

The Arrhenius equation can replace the Monod equation owing to its similar form. This study used Arrhenius equation to describe biokinetics by replacing each factor of Monod equation [29]. The developed Arrhenius equation for microbial growth and production is as follows (Equations (15) and (16)).

$$\frac{dX}{dt} = \frac{F_{\text{freq}}}{F_{\text{div}}} e^{R(T_a)} \prod_{i=1}^{n_c} c_{i,j} \tag{15}$$

with

$$F_{\text{div}} = \left(1 + Ax_{i,j}\right)^B \tag{16}$$

where $F_{\text{freq}}$ is frequency factor [l/d], $F_{\text{div}}$ is division factor [dimensionless], $R(T_a)$ is temperature-dependent reaction factor [dimensionless], $n_c$ is total number of components [dimensionless], $c_i$ is concentration of component $i$ [gmol/L], $A$ and $B$ are division factor constants [dimensionless], and $x_{i,j}$ is molar fraction of component $i$ in phase $j$ [dimensionless].

The temperature-dependent reaction rate factor reflects the effect of temperature on biokinetics as shown in Equation (17) [18,29]. It plays a role of temperature effect by replacing $f(T)$ of the Monod equation.

$$R(T_a) = \frac{-E_{\text{act}}}{RT_a} \tag{17}$$

where $E_{act}$ is activation energy [J/gmol], $R$ is gas constant [8.314 J/gmol·K], and $T_a$ is absolute temperature [K].

Because the microbial growth rate continuously changes with temperature, $R(T_a)$ should be defined with segmentation interval amounts of $A_i$ and $A_{i+1}$. Therefore, $A_i$ and $A_{i+1}$ adjusted each line segment up or down to match its neighborhood value, making $R(T_a)$ continuously. This conversion process is as follows (Equation (18))

$$A_i - \frac{E_{a,i}}{RT_{a,i}} = A_{i+1} - \frac{E_{a,i+1}}{RT_{a,i+1}} \tag{18}$$

where $A_i$ is reference segment amount of $i$th [dimensionless], $E_{a,i}$ is reference activation energy of $i$th [J/gmol], $T_{a,i}$ is reference temperature of $i$th [K], $A_{i+1}$ is reference segment amount of $i+1$th [dimensionless], $E_{a,i+1}$ is reference activation energy of $i+1$th [J/gmol], and $T_{a,i+1}$ reference temperature of $i+1$th [K].

### 2.3. Wettability Alteration by Interfacial Tension

The injected microbe undergoes metabolic activity in the aqueous phase, consuming nutrients and producing a biosurfactant. The produced biosurfactant reduces the interfacial tension between oil and water, which eventually alters relative permeability. To represent this alteration, the log of the capillary number was utilized from the following equations (Equations (19)–(21)):

$$S_j = S_{jA}(1 - r) + S_{jB}r \tag{19}$$

$$k_j = k_{jA}(1 - r) + k_{jB}r \tag{20}$$

with

$$r = \frac{\log_{10} N_c - \log_{10} N_{cA}}{\log_{10} N_{cB} - \log_{10} N_{cA}} \tag{21}$$

where $S_j$ is present saturation of phase $j$ [dimensionless], $S_{jA}$ is saturation of phase $j$ at set A [dimensionless], $S_{jB}$ is saturation of phase $j$ at set B [dimensionless], $r$ is interpolation factor [dimensionless], $k_j$ is present relative permeability of phase $j$ [dimensionless], $k_{jA}$ is relative permeability of phase $j$ at set A [dimensionless], $k_{jB}$ is relative permeability of phase $j$ at set B [dimensionless], $N_c$ is present capillary number of phase $j$ [dimensionless], $N_{cA}$ is capillary number of phase $j$ at set A [dimensionless], and $N_{cB}$ is capillary number of phase $j$ at set B [dimensionless].

The set A of relative permeability is at the low capillary number whereas set B is at the high capillary number. Using Equations (19)–(21), the interpolated relative permeability was derived based on the Brooks–Corey equation [30].

### 2.4. Mass and Energy Balance for Nonisothermal Model

Mass conservation for components was calculated as follows (Equation (22)).

$$\frac{\partial}{\partial t}\left(\varphi \rho_i \sum_{i=1}^{n_p} S_j c_{i,j}\right) + \nabla \cdot \left(\sum_{i=1}^{n_p} \rho_i c_{i,j} u_j\right) = R_i \tag{22}$$

where $\varphi$ is porosity [dimensionless], $\rho_i$ is density of component $i$ [kg/m$^3$], $n_p$ is total number of phase [dimensionless], $c_{i,j}$ is fraction of component $i$ in phase $j$ [dimensionless], $u_j$ is Darcy velocity of phase $j$ [m/s], and $R_i$ is total source/sink term for the component $i$ [kg/m$^3$·s].

Darcy velocity was defined based on Darcy's law as shown in Equation (23),

$$u_j = -\frac{k k_{rj}}{\mu_j}\nabla\left(p_j - \gamma_j h\right) \tag{23}$$



where $k$ is intrinsic permeability [md], $k_{rj}$ is relative permeability of phase $j$ [dimensionless], $\mu_j$ is viscosity of phase $j$ [cp], $p_i$ is pressure of phase $j$ [atm], $\gamma_j$ is specific weight of phase $j$ [kg/m²·s²], and $h$ is vertical depth [m].

To describe and examine the thermal effect on cold-water MEOR, the general total energy balance was used (Equation (24)) as follows:

$$\frac{\partial}{\partial t}\left[(1-\varphi)\rho_R C_{sp,R} + \varphi\sum_{i=1}^{n_p}\rho_j S_j C_{sp,j}\right]T + \nabla\cdot\left(\sum_{j=1}^{n_p}\rho_j C_{p,j}u_j T - \kappa\nabla T\right) + Q_{\text{gain}} = q_{\text{H}} \tag{24}$$

where $\rho_R$ is density of reservoir rock [kg/m³], $C_{sp,R}$ is heat capacity of rock [J/K], $\rho_j$ is density of phase $j$ [kg/m³], $C_{sp,j}$ is heat capacity of phase $j$ at constant volume [J/K], $C_{p,j}$ is heat capacity of phase $j$ at constant pressure [J/K], $\kappa$ is thermal conductivity [J/m·d·K], $Q_{\text{gain}}$ is heat gain rate [J/d], and $q_{\text{H}}$ is enthalpy source term [J/d].

Because cold water is injected into the reservoir, heat transfer occurs from the surrounding formations to the reservoir unlike a general thermal EOR. To describe the nonisothermal conditions, the heat transfer across the boundaries was considered using a semi-analytical model (Equations (19)–(21)) [31].

### 2.5. Oil Viscosity Correlation

The cold-water injection of microbial slug into the hot reservoir affects the crude oil viscosity. As increased oil viscosity causes a negative effect on oil recovery, the consideration of this change is important. Standing [32] reported the best-fit equation of the change in oil viscosity for the original Beal graphical correlation [33,34] (Equations (25) and (26)).

$$\mu_{oD} = \left(0.32 + \frac{1.8\times 10^7}{\gamma_{\text{API}}^{4.53}}\right)\left(\frac{360}{1.8(T-273.15)+232}\right)^m \tag{25}$$

with

$$m = 10^{\left(0.43 + \frac{8.33}{\gamma_{\text{API}}}\right)} \tag{26}$$

where $\mu_{oD}$ is changed oil viscosity [cp], $\gamma_{\text{API}}$ is specific API gravity [dimensionless].

The correlation was adaptable for temperature in range of 311–377 K and for gravity in range of 10–65° API.

## 3. Results and Discussion

### 3.1. Calibration of The Temperature-Dependent Reaction Factor

The effect of temperature on microbial growth rate was determined by comparing $e^{R(T_a)}$ and $f(T)$. The growth temperature conditions for *Bacillus Subtilis* are shown in Table 2 [35]. Based on the optimum, minimum, and maximum growth temperature, $r_{\max}/r_{\text{opt}}$ was plotted for the model of $f(T)$ (Figure 1).

**Table 2.** Minimum, maximum, and optimal growth temperature for *Bacillus subtilis* [35].

| Temperature Range | Value (°C) |
|---|---|
| Minimum temperature | 10.85 |
| Optimal temperature | 37.85 |
| Maximum temperature | 52.85 |

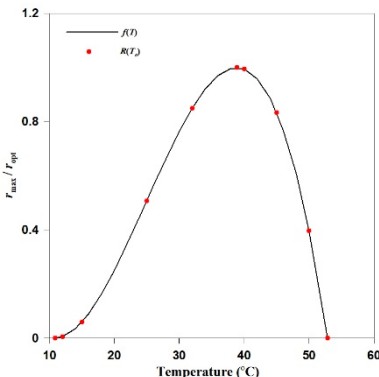

**Figure 1.** Comparison of numerical model to experimental results.

$f(T)$ was transformed into the temperature-dependent reaction factor of $e^{R(T_a)}$. Each temperature-dependent reaction factor was obtained by comparing $f(T)$. To make the nonlinear temperature-dependent reaction factor, interpolation was conducted based on segment amount of $A_i$ and $A_{i+1}$. Table 3 listed the calculated multiple activation energies which is based on the reference activation value of 52.85 °C. Figure 1 depicts the generated $e^{R(T_a)}$ with $f(T)$ to validate the effect of temperature on biokinetics.

**Table 3.** Multiple activation energies for *Bacillus subtilis*.

| Types | Temperature (°C) | Activation Energy (J/gmol) |
|---|---|---|
| derived | 10.85 | 16,324,518.55 |
| | 12 | 567,091.09 |
| | 15 | 153,312.32 |
| | 25 | 55,582.38 |
| | 32 | 18,816.06 |
| | 37.85 | −3681.10 |
| | 40 | −29,192.22 |
| | 45 | −126,491.17 |
| | 50 | −5,939,661.80 |
| reference | 52.85 | 54,881.64 |

### 3.2. Validation of Stoichiometric Reaction and Biokinetics

Based on the developed temperature-dependent reaction factor and biokinetics, history-matching was performed against the batch experiment using CMG CMOST. Conditions of the batch model for the history-matching process were obtained from the experimental data (Table 4) [14]. The electron partitions of the microbial synthesis ($f_s$) and biosurfactant production ($f_s'$) were assumed as the matching parameters to determine the stoichiometric reactions. Frequency and division factors were also regarded as matching parameters for the biokinetic model. As shown in Figure 2, the red line represents the best history-matched result from 500 simulations against the experiment. The stoichiometric reactions were obtained as Equations (27) and (28) within 2.08% of the experiment result. The obtained $f_s$, $f_s'$, $F_{freq}$, $A$, and $B$ are 0.40, 0.12, 512,000, 45,200,000, and 2, respectively. Simulations using the matched parameters indicated that *Bacillus Subtilis* growth and *Surfactin* production vary with temperature (Figure 3).

$$0.02083 \text{ Sucrose} + 0.02 \text{ NH}_4{}^+ + 0.0341 \text{ HCO}_3{}^- \overset{Bacillus\ subtilis}{\rightarrow}$$
$$0.02 Bacillus\ Subtilis + 0.014076 \text{ Fructose} + 0.007050 \text{ Lactate} + 0.007050 \text{ Acetate}$$
$$+ 0.01408 \text{ Mannitol} + 0.007050 \text{ Ethanol} + 0.041852 \text{ CO}_2 + 0.072466 \text{ H}_2\text{O} \tag{27}$$

$$0.02083 \text{ Sucrose} + 0.00289\text{NH}_4{}^+ + 0.20723 \text{ HCO}_3{}^- \xrightarrow{Bacillus\ subtilis}$$
$$0.00043\ Surfactin + 0.02064 \text{ Fructose} + 0.01034 \text{ Lactate} + 0.01034 \text{ Acetate} \tag{28}$$
$$+0.00207 \text{ Mannitol} + 0.01034 \text{ Ethanol} + 0.03944 \text{ CO}_2 + 0.00954 \text{ H}_2\text{O}$$

**Table 4.** Batch model conditions [14].

| Property | Value |
|---|---|
| Volume (L) | 1 |
| Pressure (atm) | 1 |
| Temperature (°C) | 45 |
| Initial sucrose (g/L) | 20.0 |
| Initial $NH_4{}^+$ (g/L) | 3.0 |
| Initial $HCO_3{}^-$ (g/L) | 2.8 |
| Initial microbes (g/L) | 0.1 |

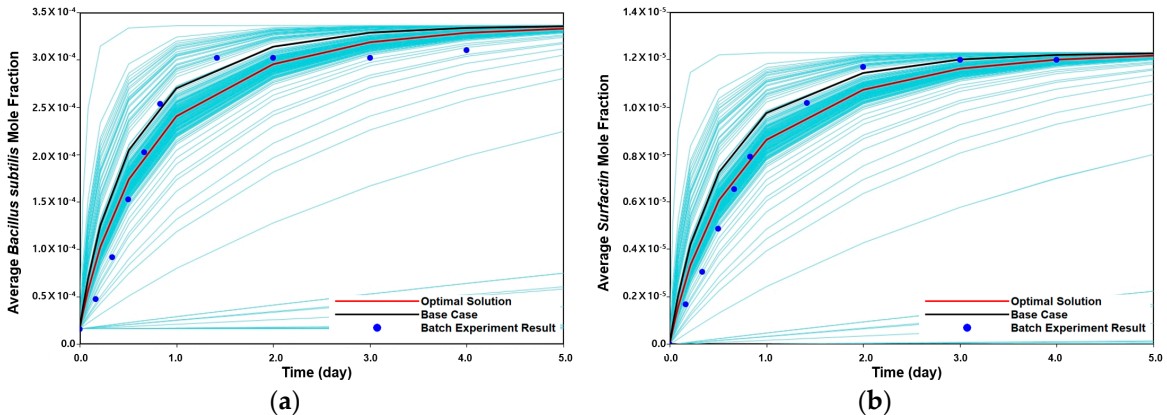

**Figure 2.** History-matched results: (**a**) *Bacillus subtilis*; (**b**) *Surfactin*.

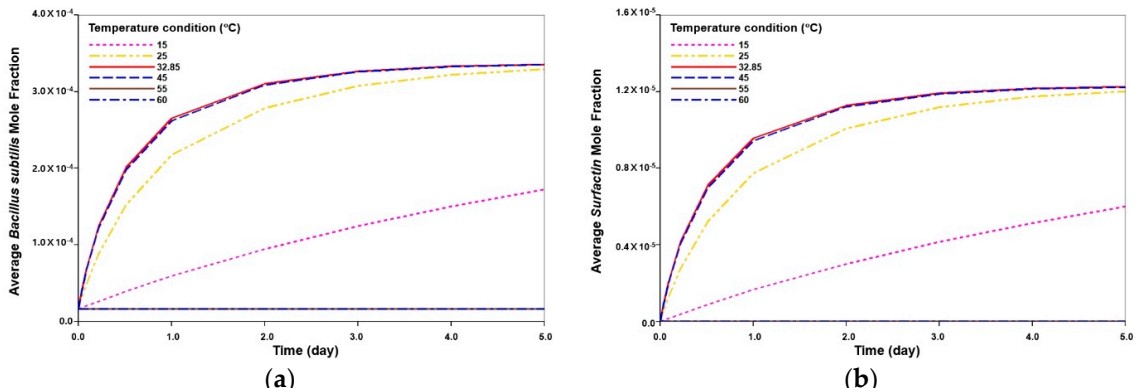

**Figure 3.** Temperature effects on microbial growth: (**a**) *Bacillus subtilis*; (**b**) *Surfactin*.

### 3.3. Parametric Study of Injection Design

#### 3.3.1. Model Description

Based on the part of reservoir model used by Hosseininoosheri et al. [7], a two-dimensional (2D) hypothetical homogenous model was constructed. It was discretized into 20 × 20 × 1 grid blocks with dimensions of 3.43 × 3.43 × 5.49 m (Figure 4). The initial conditions of the reservoir are shown in Table 5. The reservoir was assumed to be sandstone with a general rock heat capacity and conductivity [36]. The porosity (0.24), permeability (30 md), and relative permeability were also obtained from Hosseininoosheri et al. [7]. To simplify the model, we assumed that there are not any

indigenous microbes in the reservoir. During cold-water MEOR, the relative permeability was changed by the produced biosurfactant whereas the porosity (0.24) and permeability (30 md) were assumed to be constant. The change in the interfacial tension with *Surfactin* concentration was acquired from experimental data [8]. As shown in Figure 5, interfacial tension sharply decreases to 2 mN·m$^{-1}$ until *Surfactin* concentration of 44 mg·L$^{-1}$ and remains nearly constant thereafter. The relative permeability varies with capillary number which is changed via interfacial tension (Table 6) [7].

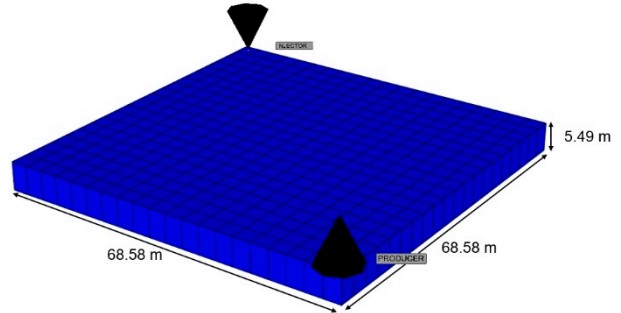

**Figure 4.** Homogeneous 2D reservoir model [7].

**Table 5.** Reservoir properties [7,36].

| Reservoir Property | Value |
| --- | --- |
| Number of grid blocks | 20 × 20 × 1 |
| Thickness (m) | 5.49 |
| Grid block size (m) | 3.43 × 3.43 × 3.43 |
| Porosity | 0.24 |
| Permeability (md) | 30 |
| Initial water saturation | 0.23 |
| Initial oil saturation | 0.77 |
| Oil compressibility (1/kPa) | $4.35 \times 10^{-7}$ |
| Rock compressibility (1/kPa) | $1.45 \times 10^{-7}$ |
| Initial reservoir pressure (kPa) | 38,611 |
| Initial reservoir temperature (°C) | 71 |
| Initial interfacial tension (mN/m) | 40 |
| Rock heat capacity (J/m$^{-3}$·°C) | 2,511,000 |
| Rock heat conductivity (J/m·d·°C) | 258,109.29 |

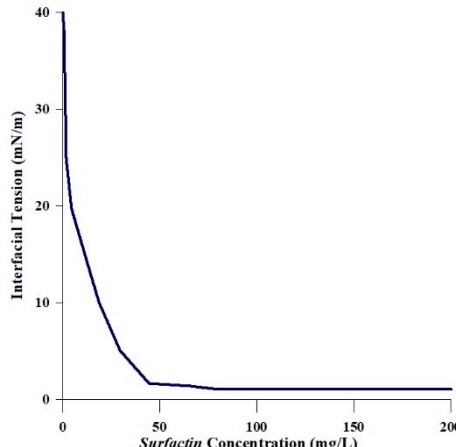

**Figure 5.** Change in interfacial tension with biosurfactant concentration [9].

**Table 6.** Relative permeability parameters for interpolation [7].

| Parameter | Phase | Capillary Number | Value |
|---|---|---|---|
| Residual saturation | Water ($S_{rw}$) | Low | 0.23 |
| | | High | 0 |
| | Oil ($S_{ro}$) | Low | 0.39 |
| | | High | 0 |
| Relative permeability at residual saturation | Water ($k_{rw}$) | Low | 0.138 |
| | | High | 1 |
| | Oil ($k_{ro}$) | Low | 0.515 |
| | | High | 1 |
| Exponent for calculating relative permeability | Water ($n_w$) | Low | 3.182 |
| | | High | 1 |
| | Oil ($n_o$) | Low | 2.405 |
| | | High | 1 |

Three stages of injection were implemented by injecting a total of 1 pore volume (PV). First waterflooding (WF) of 0.48 PV was implemented to establish the proper temperature environment, followed by MEOR for 0.02 PV. During MEOR, *Surfactin* was produced by injected nutrients and microbes. Second waterflooding of 0.5 PV was then performed to propagate the produced *Surfactin* farther inside the reservoir [6].

Injection designs can influence MEOR performance by changing the amount and distribution of produced *Surfactin*. To investigate the effect of injection design, a case study was conducted by selecting injection nutrient concentration, rate, and temperature as the injection parameters. Table 7 summarizes the injection design for the case study. To examine only the effects of nutrient concentration, microbes were injected at a constant concentration of 0.1 g/L regardless of the determined nutrient concentration.

**Table 7.** Nutrient concentration, rate, and temperature data for the case study.

| Case | Stage | Injection Temperature (°C) | Injection Rate (m³/d) | Injected Sucrose (g/L) |
|---|---|---|---|---|
| Injection nutrient concentration | 1st and 2nd WF | 35 | 6.2 | 0.0 |
| | MEOR | 35 | 6.2 | 5 |
| | | | | 20 |
| | | | | 40 |
| Injection rate | 1st and 2nd WF | 35 | 6.2 | 0 |
| | MEOR | 35 | 1.0 | 20 |
| | | | 3.1 | |
| | | | 6.2 | |
| | | | 24.8 | |
| Injection temperature | 1st and 2nd WF | Same as MEOR | 6.2 | 0 |
| | MEOR | 5 | 6.2 | 20 |
| | | 20 | | |
| | | 35 | | |
| | | 45 | | |

### 3.3.2. Analysis of Injection Parameters

To evaluate the optimized performance of cold-water MEOR, a series of changes in injection conditions were simulated and compared. MEOR was conducted by injecting 0.02 PV of nutrients and microbes.

To examine the effect of nutrient concentration on oil recovery, five scenarios were implemented by changing the sucrose concentration from 5 to 40 g/L. Injection temperature and rate were fixed to 35 °C and 6.2 m³/d, respectively. Figure 6 depicts the history of oil recovery with injected sucrose concentration. A higher sucrose concentration resulted in more *Surfactin* production, thereby reducing

interfacial tension to increase oil recovery (Figure 7). A sucrose concentration of 40 g/L enhanced oil recovery up to 51% (Figure 6). The decreased *Surfactin* concentration in Figure 7 represented the effect of second waterflooding. The second waterflooding following MERO caused *Surfactin* to propagate in the reservoir thereby increasing oil recovery.

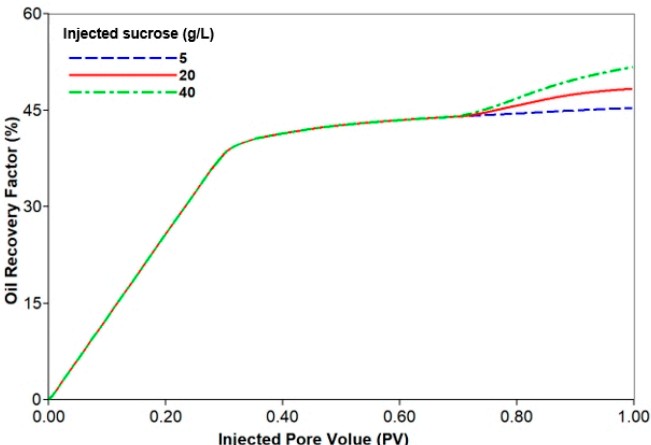

**Figure 6.** Oil recovery factors with various injected nutrient concentrations.

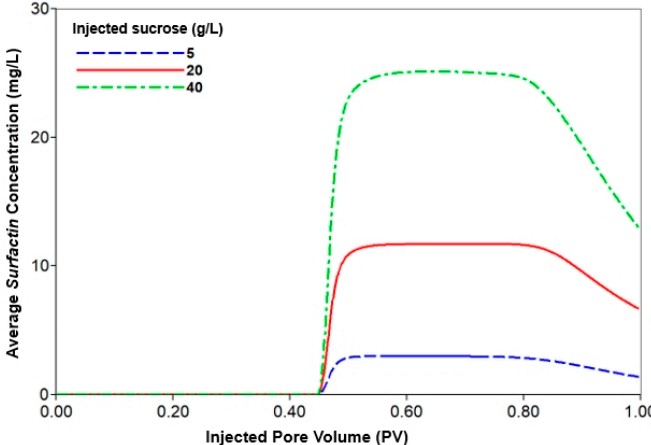

**Figure 7.** Average *Surfactin* concentration with various injected nutrient concentrations.

To examine the effect of injection rate on oil recovery, a constant concentration of nutrients and microbes was injected at 35 °C. The injection rate was adjusted from 1.0 to 24.8 m³/d and the injected PV for MEOR was set to 0.02. Therefore, a longer period of MEOR was conducted at a lower injection rate. Figure 8 illustrates a maximum recovery of 50% using the injection rate of 1.0 m³/d. The change in injection rate affected not only heat transfer but also nutrient residence time to react. A higher injection rate resulted in less production of *Surfactin* as shown in Figure 9. To investigate the reasons for this, the average produced *Surfactin* concentration and temperature were compared for the rates of 1.0 and 24.8 m³/d (Figures 9 and 10).

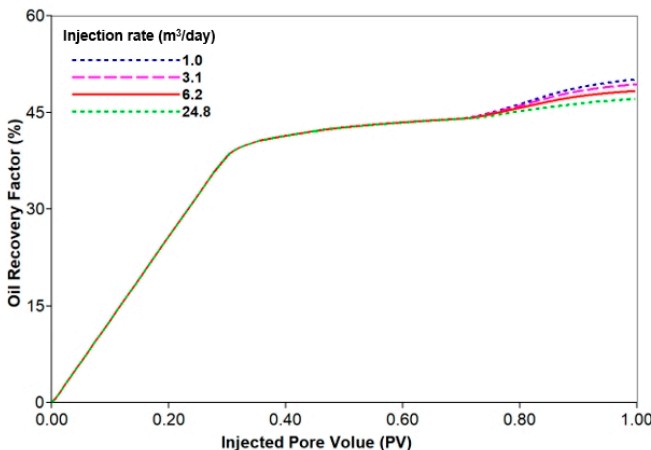

**Figure 8.** Oil recovery factors with various injection rates.

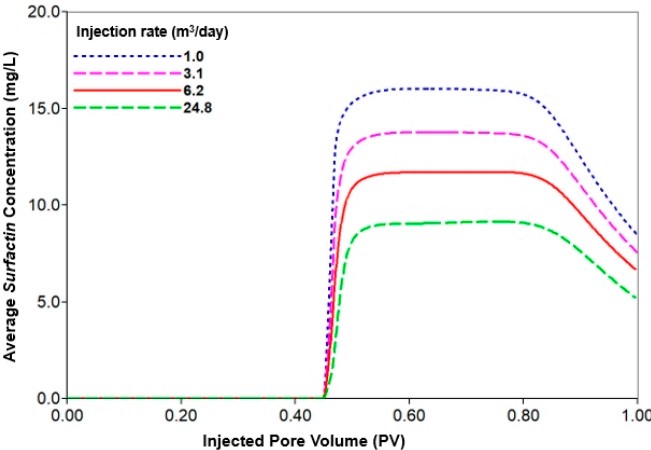

**Figure 9.** Average *Surfactin* concentration with various injection rates.

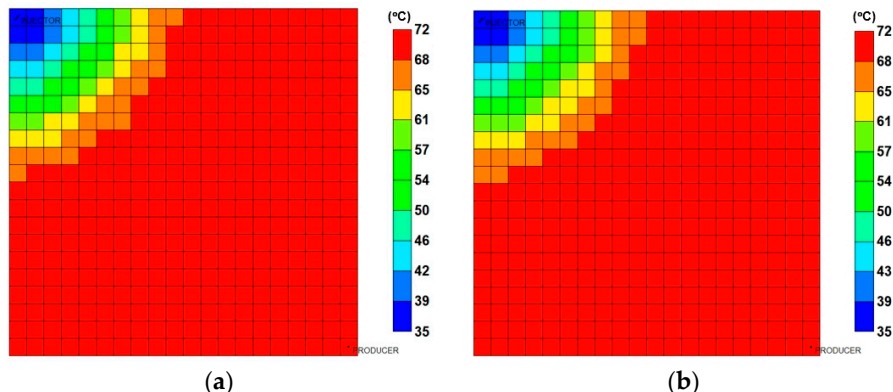

**Figure 10.** Temperature distribution at injection rates: (**a**) 1.0 m³/d; (**b**) 24.8 m³/d.

Figures 10 and 11 show the distribution of temperature and interfacial tension, respectively. The results showed that cold-water MEOR efficiency correlated with temperature and nutrient residence time. Because the higher injection rate of 24.8 m³/d transferred more cold water deep inside the reservoir, a more hospitable temperature environment for microbial growth was identified (Figure 10). However, the produced *Surfactin* concentration at 24.8 m³/d was lower than that at 1.0 m³/d as the injected nutrients were swept out before *Surfactin* production. This eventually had a negative effect on the distribution of interfacial tension (Figure 11). The interfacial tension of 40 mN/m near the injection well was a phenomenon caused by the second waterflooding.

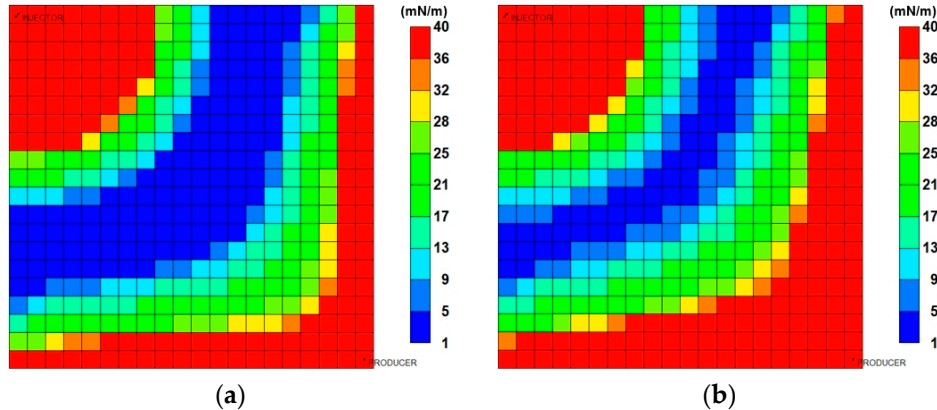

**Figure 11.** Interfacial tension distribution at injection rates: (**a**) 1.0 m³/d; (**b**) 24.8 m³/d.

Via cold-water injection, microbial growth occurred even in the high-temperature reservoir. The temperature of the reservoir varied from the initial temperature (71 °C) to the injection temperature. The effect of the injection temperature was examined by adjusting the value from 5 to 45 °C. As shown in Figure 12, the injection temperature of 20 °C resulted in the best improvement of 6% in oil recovery compared to waterflooding. Figure 13 shows the effect of injection temperature on oil recovery by plotting the average produced *Surfactin* concentration. The injection temperature of 20 °C resulted in the maximum produced *Surfactin* concentration compared to that of the other cases.

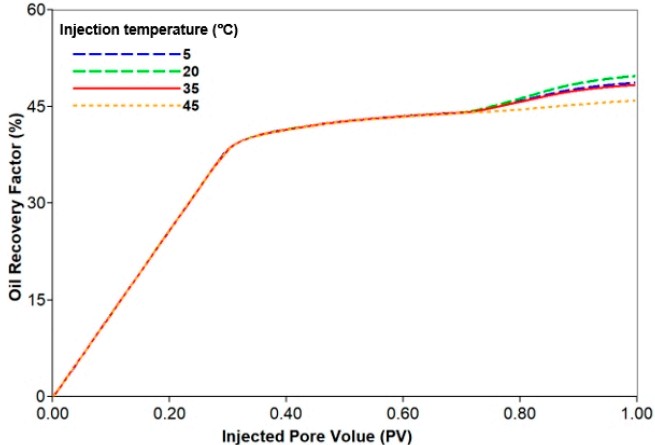

**Figure 12.** Oil recovery factors with various injection temperatures.

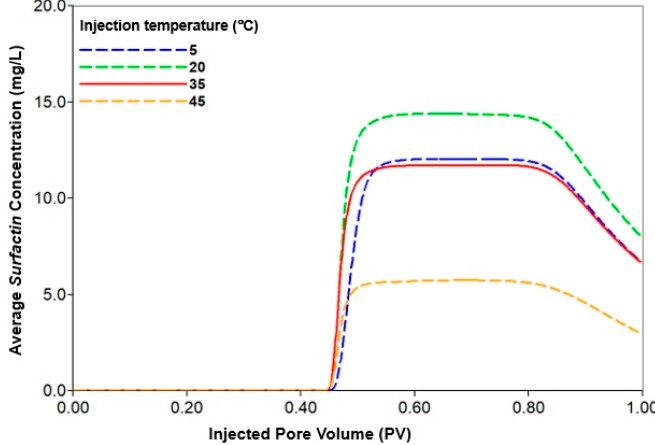

**Figure 13.** Average *Surfactin* concentration with various injection temperatures.

Further examination was conducted by comparing injection temperatures of 20 °C and 45 °C. The distributions of temperature and interfacial tension are illustrated in Figures 14 and 15. The results showed that injection temperature could significantly affect oil recovery by changing the reservoir temperature. An injection temperature of 20 °C resulted in a more hospitable temperature condition for microbial growth than that at 45 °C (Figure 14). The more hospitable environment for microbial growth led to a higher *Surfactin* concentration as shown in Figure 13. Consequently, the distribution of interfacial tension at an injection temperature of 20 °C was more favorable than that at 45 °C (Figure 15).

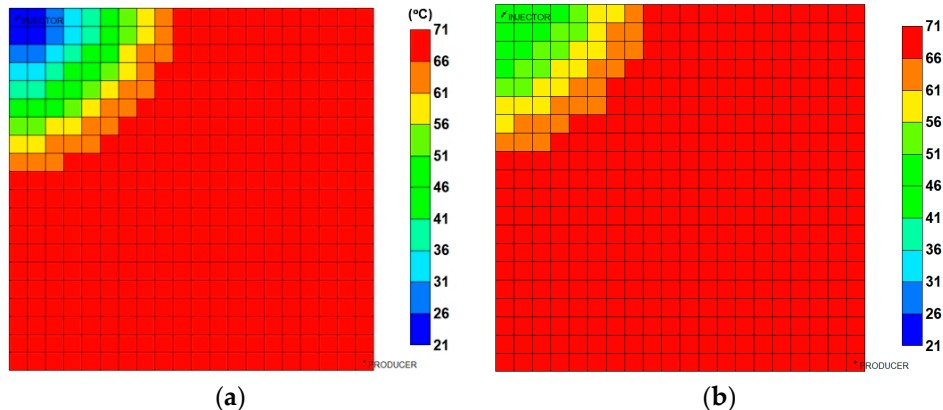

**Figure 14.** Temperature distribution at injection temperatures: (**a**) 20 °C; (**b**) 45 °C.

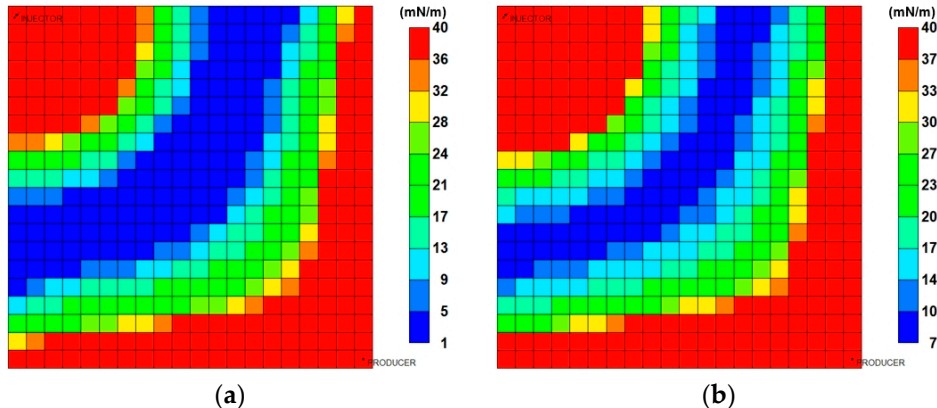

**Figure 15.** Interfacial tension distribution at injection temperatures: (**a**) 20 °C; (**b**) 45 °C.

### 3.3.3. Parametric Analysis and Optimization Process

To investigate the optimal injection design, parameter analysis and optimization process to maximize oil recovery were conducted. Nutrient concentration, rate, and temperature were adjusted as injection design parameters. By injecting a total of 1 PV, the sequences of injection followed the three stages of case study. The parametric analysis was implemented by changing the injection parameters independently. The ranges of the injection parameters for the parametric analysis are summarized in Table 7. Via parameter analysis, each injection factors showed a correlation with the oil recovery factor.

The oil recovery factor was positively correlated with the injected sucrose concentration as shown in Figure 16. A higher sucrose concentration produced more *Surfactin* to reduce interfacial tension, which eventually increased oil recovery. Because the interfacial tension no longer decreased at a *Surfactin* concentration higher than 44 mg/L, MEOR efficiency improved little at a sucrose concentration higher than 50 g/L (Figures 5 and 16).

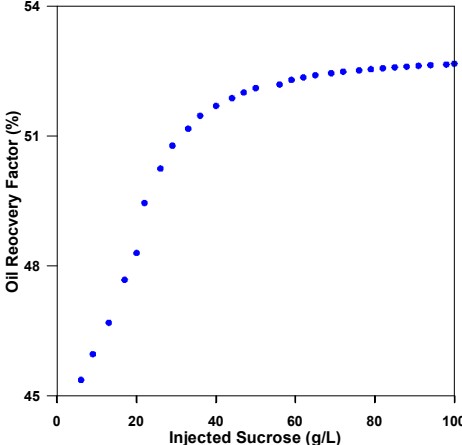

**Figure 16.** Relationship between sucrose concentration and oil recovery factor.

Figure 17 shows the correlation between the injection rate and oil recovery factor. Because the injected PV for MEOR was set to 0.02, the injection period of MEOR changed. An injection rate that was too low (<1.0 m³/d) could not create a favorable temperature for the microbes to produce *Surfactin*, which eventually had a negative impact on oil recovery. Therefore, the oil recovery factor was positively correlated with this range of injection rate (<1.0 m³/d). When the injection rate was higher than 1.0 m³/d, a negative correlation was found because the period of MEOR was too short to produce *Surfactin*. This result indicated that the oil recovery factor was more affected by the period of MEOR than the temperature environment. Thus, the effect of the injection rate on oil recovery suggested that it had an optimal value by adjusting the nutrient residence time and heat transfer of the oil recovery factor.

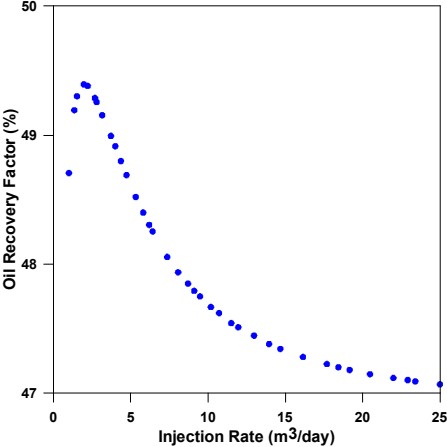

**Figure 17.** Relationship between injection rate and oil recovery factor.

In the case of injection temperature, the oil recovery factor was correlated with injection temperature as parabolic curves (Figure 18). An injection temperature lower than 21 °C decreased the improvement of oil recovery as the lower *Surfactin* concentration was generated by inappropriate temperature condition. However, oil recovery factor was no longer decreased with an injection temperature lower than 10 °C. With this injection temperature range (<10 °C), the cold water slowly heated up to favorable growth temperature condition and produce more *Surfactin* in high temperature reservoir. Therefore, *Bacillus subtilis* growth has become active in deep inside the reservoir, thereby producing more *Surfactin*. On the other hand, an injection temperatures higher than 21 °C generated an unfavorable environment for *Surfactin* production again according to the survival range of the microbes. That is, the oil recovery factor dramatically decreased within this injection temperature

range (>21 °C). Through parametric analysis, the independent impact of each injection parameter on the oil recovery factor was identified. As the results, optimization of the injection design via adjustment of these parameters is essential.

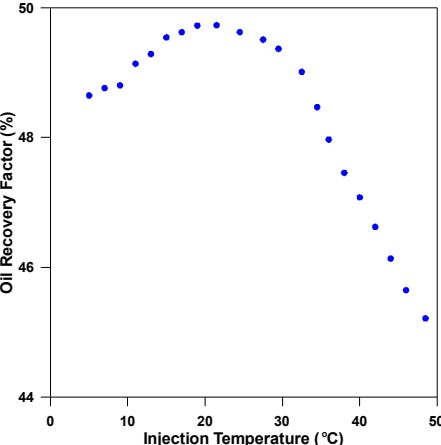

**Figure 18.** Relationship between injection temperature and oil recovery factor.

Finally, an optimization process was conducted by simultaneously adjusting all injection parameters. The range of each injection parameter for MEOR is listed in Table 8. The three stages of the case study were also utilized to obtain an optimal injection design. The first waterflooding of 0.48 PV, the MEOR of 0.02 PV, and the second waterflooding of 0.5 PV were conducted in sequence. Therefore, the period of MEOR was adjusted with the injection rate, thereby injecting 0.02 PV. The base case was designed as a sucrose concentration of 20 g/L, an injection rate of 6.2 m$^3$/d, and a temperature of 35 °C. Three-hundred simulations were conducted to determine the optimal injection design solution (Figure 19). The results suggested that a sucrose concentration of 100 g/L, an injection rate of 7 m$^3$/d, and a temperature of 19 °C were the optimal injection parameters. Via the optimization process, the maximum oil recovery factor of 53% was achieved. It was comparable not only to waterflooding (45%) but also to the base case (47%).

**Table 8.** Injection parameter ranges for the sensitivity analysis and optimization process.

| Parameter | Range |
|---|---|
| Injected sucrose and microbe concentration (g/L) | 1st and 2nd WF: sucrose 0 + microbe 0<br>MEOR: sucrose 5–40+microbe 0.1<br>(Base case MEOR: sucrose 20+microbe 0.1) |
| Injection Rate (m$^3$/d) | 1st and 2nd WF: 6.2<br>MEOR: 1–25<br>(Base case MEOR: 6.2) |
| Injection Temperature (°C) | 1st and 2nd WF: same as MEOR<br>MEOR: 5–50<br>(Base case MEOR: 35) |

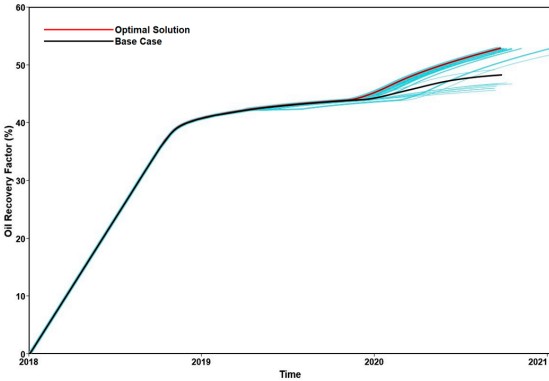

**Figure 19.** Optimization results.

## 4. Conclusions

This study evaluated the performance of cold-water MEOR in a high-temperature reservoir. Based on a coupled biokinetic and nonisothermal multiphase model, a numerical model was developed to characterize the *Surfactin* effect. Validation and calibration were conducted to obtain stoichiometric reactions and biokinetic parameters against experimental results.

Further examinations were implemented to identify the optimal injection design for cold-water MEOR. A higher sucrose concentration generated more *Surfactin*, which eventually affected the oil recovery factor. A higher injection rate resulted in a more favorable temperature condition for growth, but it swept out the microbes prior to *Surfactin* production. On the other hand, a very low injection rate resulted in an unfavorable condition of heat transfer. Thus, the injection rate had to be optimized, thereby satisfying both an appropriate heat transfer and nutrient residence time. Injection temperature was related not only to *Surfactin* production but also oil viscosity by changing reservoir conditions. According to the microbial survival temperature range, injected cold water induced thriving microbes. However, a too-low injection temperature resulted in an increase in oil viscosity, thereby reducing the MEOR efficiency.

The correlation between each injection parameter (sucrose concentration, injection rate, and temperature) and the oil recovery factor indicated that each parameter could have an impact on oil recovery efficiency. The injection design was optimized by simultaneously adjusting all injection parameters. Compared to the base case (a sucrose concentration of 20 g/L, an injection rate of 6.2 m$^3$/d, and a temperature of 35 °C), a 6% higher oil recovery was accomplished using the optimal injection design (a sucrose concentration of 100 g/L, a rate of 7 m$^3$/d, and a temperature of 19 °C). Through this study, the potential of MEOR in a high-temperature reservoir has been identified using the developed model.

**Author Contributions:** Conceptualization and methodology, E.H. and K.S.L.; software and validation, E.H., J.H.C. and M.S.J.; formal analysis and investigation, E.H. and T.H.K.; resources, data curation, and visualization; E.H. and J.H.L.; writing—original draft preparation, E.H.; writing—review and editing, J.H.C. and K.S.L.; supervision, project administration, and funding acquisition, K.S.L.

**Funding:** This research was funded by the project "Development of IOR/EOR Technologies and Field Verification for Carbonate Reservoir in UAE" through the Korean Government Ministry of Trade, grant number 20152510101980.

**Conflicts of Interest:** The authors declare no conflicts of interest.

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
