# Peer review of "Development of Coupled Biokinetic and Thermal Model to Optimize Cold-Water Microbial Enhanced Oil Recovery (MEOR) in Homogenous Reservoir"

_sustainability, doi:10.3390/su11061652_

Reviewer 1 Report

General comments

The study has provided a comprehensive biokinetic models, which could sufficiently be used to provide clearer quantification of surfactant concentration and IFT reduction despite the complexity of such model development. Again, the stoichiometric models developed were in fact very good contribution. However, the models developed and results obtained need to be compared with experimental data to establish the basis for oil field implementation. This can be achieved by establishing separate graphs followed by discussions on such plots. Particularly, where it was noticed that recovery factor was significantly affected by period of injection, subsequent schemes developed through thus study can be validated to ensure comparable agreement with experimental studies.

Specific comments:

Line 12: Bacillus subtilis can be a good candidate but require justification here, when investigated in other study, it had least IFT reduction  compared to  licheniformis, Paneabacillus Polymyxa etc, check this http://www.joetsite.com/wp-content/uploads/2017/07/Vol.-62-47-2017.pdf

Line 13: “…..against experimental results” need to support this by comprehensive comparison using published studies under similar conditions.

Line 34: “Previous studies….”. have these studies used experimental studies with bacillus subtilis?

Line 43: thermophilic bacillus needs to be argued here to support the higher temperature application

Line 68-69: wettability modification of surfactin to be replaced with wettability modification by surfactin

Line 219, Table 2, values of the 32oC achieved could be supported through other experimental studies that found such range of bacillus growth at 37C.

Line 255:… interfacial tension sharply decreases to 44 mg.L” The sentence need modification as 44mg/L is not IFT value but rather surfactin conc.

Line 384: grammar check required

Author Response

General comments

The study has provided a comprehensive biokinetic models, which could sufficiently be used to provide clearer quantification of surfactant concentration and IFT reduction despite the complexity of such model development. Again, the stoichiometric models developed were in fact very good contribution. However, the models developed and results obtained need to be compared with experimental data to establish the basis for oil field implementation. This can be achieved by establishing separate graphs followed by discussions on such plots. Particularly, where it was noticed that recovery factor was significantly affected by period of injection, subsequent schemes developed through thus study can be validated to ensure comparable agreement with experimental studies.

Specific comments

Point 1 (Line 12): Bacillus subtilis can be a good candidate but require justification here, when investigated in other study, it had least IFT reduction  compared to  licheniformis,Paneabacillus Polymyxa etc, check this http://www.joetsite.com/wp-content/uploads/2017/07/Vol.-62-47-2017.pdf

Response 1: We also considered microbes like Bacillus licheniformis, Paneabacillus Polymyxa etc., which have a greater IFT reduction than Bacillus subtilis. However, Bacillus subtilis is much easier to use for petroleum engineering applications because it is the most-frequently mentioned microbial species with a clear growth experiment. In addition, its degree of IFT reduction is enough to increase oil recovery in reservoir condition.

Point 2 (Line 13): “…..against experimental results” need to support this by comprehensive comparison using published studies under similar conditions.

Response 2: We calibrated the experimental results of Makker et al. (1997) which is mentioned in Line 230 and Table 4. In Line 231, the citation was changed to reference 14 to correct the wrong reference number. Figure 2 showed the matched-results with experiment for Bacillus subtilis and produced Surfatin. All the growth condition and growth trends of Bacillus subtilis were obtained from the results of Makkar et al.’s paper. In Figure 3, we examined growth tendency by changing the temperature according to the matched-parameters from Figure 2.

Point 3 (Line 34): “Previous studies….”. have these studies used experimental studies with bacillus subtilis?

Response 3: The referenced papers described the general aspects and potential of MEOR processes. These studies indicated that MEOR has shown sufficient potential to improve oil recovery.

Point 4 (Line 43): thermophilic bacillus needs to be argued here to support the higher temperature application

Response 4: Since this research focuses on the modelling of a cold water injection associated with MEOR, the application of thermophilic bacillus is not within the scope of this study. Following reviwer’s suggestion, we added brief comments as “Even with thermophilic microbes, the effect is small because the microbial growth rate has parabolic relationship with temperature.”.

Point 5(Line 68-69): wettability modification of surfactin to be replaced with wettability modification by surfactin

Response 5: In Line 70, we correct the preposition by replacing “of” to “by”.

Point 6 (Line 219, Table 2): values of the 32oC achieved could be supported through other experimental studies that found such range of bacillus growth at 37oC.

Response 6: The temperature value of 32oC was misspelled and corrected to 37 oC, which is also shown in Figure 1.

Point 7 (Line 255):”… interfacial tension sharply decreases to 44 mg.L” The sentence need modification as 44mg/L is not IFT value but rather surfactin conc.

Response 7: In Line 256, we correct the sentence as “interfacial tension sharply decreases to 2 mN/m until Surfactin concentration of 44 mg/L”.

Point 8 (Line 384): grammar check required

Response 8: We checked grammar and corrected it.

Reviewer 2 Report

The paper was presented in a systematic way that is easily to understand. A couple of improvement should be applied

minor miss spell %p at abstract, page 13, and conclusion part

The model only consider homogeneous reservoir, therefore it should be mentioned in the title. If applied in heterogeneous reservoir, the result might be significantly different as fingering might occur

It should be stated in the model that the author consider no partitioning in the produced surfactin (in contrast to citation no 1 by Nielsen who consider partitioning in the surfactant), which later on support by lab result by citation 11 Halim et. al

 Page 9, line 251-252. The statement "because of high temperature of reservoir .... indigenous microbes did not exist" is not supported by any proof. In fact most of the researchers in this area, agree that indigenous microbes exist in all reservoir. Recent advances in molecular biotechnology revealed thermophilic and hyperthermophilic microorganism (bacteria, methanogens and archaea) exist in oil reservoirs. This statement should be removed from the paper. The author can mentioned, "to simplified the model, we consider the indigenous microbes do not exist" or something similar to this statement.

Page 9, line 255. "The interfacial tension decreases to 44 mg L-1" is wrong. Does not match with data on Fig 5. Fig 5 shows the IFT decreases to near 0 from 40 mN/m. In addition, IFT need to be reduced into several order of magnitude to be able to get a significant results (Gray et al 2008, Willhite and Green, 1998, Stalkup 1984, Armstrong, 2012). I believe in case of MEOR, it is contribution of multiple actions simultaneously. As mentioned in the equation, there are organic acids (lactate, acetate) and solvent (ethanol) produced in the system. These other bioproducts also contribute to the increase of the oil recovery. 

Figure 7, the author need to explain why increasing nutrient concentration from 5, 20,40 g/L decrease the surfactin concentration at higher PV. This especially can be clearly seen at 40 g/L, where the surfactin concentration significantly decrease after 0.8 PV. Surprisingly, figure 6 shows oil recovery increase at 0.8 PV when the surfactin concentration decreases. This to me indicates, the surfactin might have migrate to the oil phase (partition) and not capture/detected in the water phase. As a matter of fact on the whole figure (including varying the temperature and injection rate) there's seems to be a cut off at 0.8PV. 

Author Response

General comments

The paper was presented in a systematic way that is easily to understand. A couple of improvement should be applied

Specific comments

Point 1: minor miss spell %p at abstract, page 13, and conclusion part

Response 1: We checked and corrected them.

Point 2: The model only consider homogeneous reservoir, therefore it should be mentioned in the title. If applied in heterogeneous reservoir, the result might be significantly different as fingering might occur

Response 2: We changed the title as “Development of Coupled Biokinetic and Thermal Model to Optimize Cold-Water Microbial Enhanced Oil Recovery (MEOR) in Homogeneous Reservoir”. As the reviewer mentioned, biosurfactant is not appropriate for heterogeneous reservoirs due to its lack of capability to control mobility ratio. In heterogeneous reservoirs, the application of biopolymer is recommended.

Point 3: It should be stated in the model that the author consider no partitioning in the produced Surfactin (in contrast to citation no 1 by Nielsen who consider partitioning in the surfactant), which later on support by lab result by citation 11 Halim et.al

Response 3: We mentioned about the point in page 2, line 69 as “In addition, partitioning of the produced Surfacitn was not considered to simplify the model.”.

Point 4:  Page 9, line 251-252. The statement "because of high temperature of reservoir .... indigenous microbes did not exist" is not supported by any proof. In fact most of the researchers in this area, agree that indigenous microbes exist in all reservoir. Recent advances in molecular biotechnology revealed thermophilic and hyperthermophilic microorganism (bacteria, methanogens and archaea) exist in oil reservoirs. This statement should be removed from the paper. The author can mentioned, "to simplified the model, we consider the indigenous microbes do not exist" or something similar to this statement.

Response 4: Page 9, line 253-254. Following the review’s commnts, we changed the sentence as “To simplify the model, we assumed that there are not any indigenous microbes in the reservoir”.

Point 5: Figure 7, the author need to explain why increasing nutrient concentration from 5, 20,40 g/L decrease the Surfactin concentration at higher PV. This especially can be clearly seen at 40 g/L, where the Surfactin concentration significantly decrease after 0.8 PV. Surprisingly, figure 6 shows oil recovery increase at 0.8 PV when the Surfactin concentration decreases. This to me indicates, the Surfactin might have migrate to the oil phase (partition) and not capture/detected in the water phase. As a matter of fact on the whole figure (including varying the temperature and injection rate) there's seems to be a cut off at 0.8PV. 

Response 5: Since the partitioning effect was not considered in our model, there is no migration of Surfactin between oil and water phase. The decrease of Surfactin concentration is caused by the second waterflooding. The second waterflooding after MEOR, swept the produced Surfacitn away. Therefore, Surfactin concentration decreases in the reservoir after 0.8 PV injection. To avoid the confusion, we added the sentence as “The decreased Surfactin concentration in Figure 7 represented the effect of second waterflooding. The second waterflooding following MEOR caused Surfactin to propagate in the reservoir thereby increasing oil recovery.” in Page 10, line 286.
